# Social support and health related quality of life among older people in covid-19 pandemic: The mediating role of resilience

**Badriyeh Karami**[1], **Shahab Rezaeian**[2,3], **Mansour Bagherinia**[4], **Ebrahim Shakiba**[5,6], **Raheleh Maleki**[7], **Mohammadrasool Ghasemianrad**[8], **Amirhossien Naghibzadeh**[8], **Hamzeh Zahabi**[8], **Hadi Darvishigilan**[9]*

1 Behavioral Diseases Research Center, Health Institute, Kermanshah University of Medical Sciences, Kermanshah, Iran, 2 Infectious Diseases Research Center, Health Institute, Kermanshah University of Medical Sciences, Kermanshah, Iran, 3 Department of Epidemiology, School of Health, Kermanshah University of Medical Sciences, Kermanshah, Iran, 4 School of Health, Ilam University of Medical sciences, Ilam, Iran, 5 Department of Biochemistry, School of Medicine, Kermanshah University of Medical Sciences, Kermanshah, Iran, 6 Behavioral Diseases Research Center, Kermanshah University of Medical Sciences, Kermanshah, Iran, 7 Department of Healthcare Services Management, School of Health, Ahvaz Jundishapur University of Medical Sciences, Ahvaz, Khuzestan, Iran, 8 Student Research Committee, Kermanshah University of Medical Sciences, Kermanshah, Iran, 9 Social Determinants of Health Research Center, Ahvaz Jundishapur University of Medical Sciences, Ahvaz, Iran

* hadidarvishi2000@yahoo.com

## Abstract

### Background

During emergencies and crises, comprehending the influence of various factors on the health-related quality of life (HRQoL) of older adults is crucial for enhancing it. This study aimed to investigate the mediating role of resilience in the relationship between social support and health related quality of life.

### Methods

In this descriptive-analytical cross-sectional study, 341 older adults over 60 years participated, in 2023. Vaux's social support questionnaire, SF-36 health-related quality of life questionnaire, and Conner and Davidson's resilience questionnaire were used. Structural equation modeling (SEM) was used to determine the role of resilience as a mediator between social support and HRQoL. Statistical analysis of data was performed using STATA 14.2 software.

### Results

Based on the results, a one-unit increase in social support is associated with a 0.44-unit increase in resilience ($\beta = 0.44$) and a 0.24-unit increase in HRQoL ($\beta = 0.24$) on average. Furthermore, considering the mediating role of resilience, a one-unit increase in social support contributes 0.097-units directly ($\beta = 0.097$) and 0.12-units

**Data availability statement:** All relevant data are within the paper and its Supporting Information files.

**Funding:** The author(s) received no specific funding for this work.

**Competing interests:** The authors declare that they have no competing interests.

**Abbreviations:** WHO, World Health Organization; HRQOL, health-related quality of life; QOL, quality of life; SEM, structural equation model; SE, standard error.

indirectly ($\beta = 0.12$) to HRQoL on average. The total effect of social support on HRQoL was also significant ($\beta = 0.22$).

## Conclusion

Considering the mediating role of resilience in the relationship between social support and HRQoL among older people during the Covid-19 pandemic and the possibility of emergencies and pandemics in the future, it is necessary to carry out interventions focusing on improving resilience skills to make social support more effective in the elderly and to improve the quality of life related to health.

## Introduction

The most recent disease to be classified as a pandemic by the World Health Organization (WHO) in 2020 is Covid-19 [1]. Following the onset of epidemics, societal changes have ensued, impacting various facets of individuals' welfare [2]. While the disease can affect individuals of any age, the elderly have been identified as the most susceptible and significantly affected demographic group [3]. "This issue is particularly important for nations experiencing population aging. Iran is one of the countries with a high rate of population aging. Iran is one of the countries with a high rate of population aging [4] and is expected to see substantial increase in the proportion of its elderly population over the next three decades [5].

Various studies indicate that the measures implemented during the Covid-19 crisis have adversely affected different aspects of health and quality of life (QoL) [6–8]. On the other hand, previous research findings reveal that the QoL among elderly individuals in Iran is at a moderate level, highlighting the necessity for targeted interventions [9]. QoL includes general quality of life and health-related quality of life (HRQoL) [10]. HRQoL, influenced by one's experiences, beliefs, expectations, and personal perceptions, is rooted in health and constitutes both positive and negative health dimensions [11].

One of the key determinants of HRQoL in the elderly is social support [12,13]. Various research findings conducted among the elderly population amidst the Covid-19 pandemic have indicated a significant association and a direct link between social support and HRQoL [14–17]. Newcomb & Bentler define social support as a system of interpersonal connections that offer companionship, collaboration, emotional encouragement, and in addition to promoting health-enhancing behaviors, mitigating stressful life events, and attaining individual objectives [15].

Another determinant influencing an individual's QoL is the capacity to navigate the novel life circumstances resulting of Covid-19 pandemic, which is called resilience. Resilience is the ability of an individual to cope with and adapt to new life circumstances, challenges, or crises [18]. Empirical research has indicated that an individual's resilience can significantly impact their QoL [19,20] and exhibits a positive correlation with QoL [21–23]. Moreover, various studies have indicated that resilience training contributes to an enhancement in quality of life, coping

mechanisms, and mental health among the elderly population [19]. Utilizing structural equation modeling, Gerino et al., elucidated the mediating role of resilience in the interplay between physical and mental QoL and feelings of loneliness among older adults [20].

Enhancing the quality of life for older individuals stands as a fundamental objective for nations worldwide [24]. Consequently, it is crucial to investigate how the various issues related to both the physical and psychological dimensions of aging impact the well-being of the elderly population. The findings from global studies indicate that research focusing on the HRQoL of the elderly during the Covid-19 pandemic has primarily investigated the HRQoL status and its correlation with psychological problems, particularly among among elderly patients [8,25,26]. In Iran, limited studies have also been conducted that have examined the status of overall QoL of the elderly and individual factors affecting it [27] and HRQoL during the Covid-19 pandemic has not been focused on. According to the findings of current research, there is no study that has explored the simultaneous relationship between HRQOL, social support, and resilience in the context of the Covid-19 pandemic among older adults. Consequently, the primary objective of this study is to examine the interconnections among these variables within the elderly people.

## Materials and methods

### Study setting

This descriptive-analytical cross-sectional study was carried out, involving community-dwelling seniors aged over 60 years in Kermanshah city in the western region of Iran.

### Sample and sampling method

To determine the sample size, the correlation coefficient of 0.22, which reflects the relationship between social support and HRQoL as reported in the study by Saber and Nosratabadi [28] was employed. Utilizing the sample size formula and 0.9 power to detect an effect, it was projected that 310 older individuals would be necessary. Considering a 10% non-response rate for the study, the final estimated sample size was determined to be 341. Given the potential impact of individuals' residential location on HRQoL, the division of Kermanshah city into eight districts was done based on Sahebi et al.'s research [29]. In the previously mentioned study, the eight districts of Kermanshah city were classified into three categories: poor districts (3, 2, 8), medium districts (5, 1, 6), and good districts (4, 7) based on the livability index, which encompasses three dimensions: socio-cultural, economic, and environmental. Consequently, participants were chosen utilizing the cluster random sampling technique. Each district comprises multiple neighborhoods that differ greatly in terms of socio economic status. Therefore, according to the research team's decision, two neighborhoods were randomly chosen from each district (totally 16 neighborhoods.

The requirements for participating in the study included aged over 60 years, willing to participate in the study, not having sensory perception disorders or mental retardation, and the ability to speak and understand Persian or Kurdish.

Questionnaires:

- Vaux's social support questionnaire [30] was implemented to assess social support. This questionnaire comprises 23 items utilizing a 5-point Likert response scale." Within the research conducted by Afrashteh et al., in Iran [31], the alpha coefficient of this questionnaire was reported as 0.80. The social support scores ranges from 23 to 115, with a higher score indicating a greater level of social support.

- SF-36 health-related quality of life questionnaire [32] consists of 36 questions and encompasses 8 distinct subscales. These subscales are further categorized into 4 components related to the physical domain and 4 components related to the psychological domain. The score range for HRQoL extends from 0 to 100, with higher scores being indicative of an elevated HRQoL level. Notably, this instrument demonstrates a commendable level of reliability and validity, having been scrutinized across various nations globally [33]. In the examination conducted by Montazeri et al., the questionnaire's

reliability was assessed in Iran. The evaluation of "internal consistency" unveiled that, with the exception of the vitality subscale, the remaining components of the Persian version of SF-36 exhibit minimal standard reliability coefficients ranging from 0.77 to 0.9 [34].

- To assess resilience, Conner and Davidson's 25-item resilience questionnaire [35] was administered. The questionnaire encompasses four dimensions: goal recognition and individual control (9 items), adaptation and tolerance of individual effects (10 items), leadership and trust in instinct (action based on sense) (4 items), and spiritual knowledge of the future (2 items). This instrument employs a 5-point Likert scale with a possible score range of 25–125. Ahangarzadeh et al., conducted a study in Iran to evaluate the self-validity and reliability of this questionnaire, revealing an internal consistency of 0.82 as determined by Cronbach's alpha [36]. In addition, in Noroozi et al., [37] and Afrashteh et al., [31]studies conducted among older people, cronbach's alpha coefficient was calculated to be 0.79 and 0.89, respectively.

### Data collection

Two interviewers were trained by the research team for data collection once the ethical committee of Kermanshah University of Medical Sciences (KUMS) gave the study permission to begin (ethics code IR.KUMS.REC.1402.211). Data collection was conducted at parks and recreational areas in Kermanshah, from August 15, to November 20, 2023, by interviewers. Following their explanation of the study's objectives, participants signed a consent form and filled out the pertinent questionnaire. Trained interviewers would step in to assist if participants were incapable of completing the questionnaire due to any reasons, such as illiteracy. The process of distributing and completing questionnaires continued until all questionnaires were completed.

### Statistical analysis

Descriptive statistics were first calculated to summarize the characteristics of the study sample, including frequencies and percentages for categorical variables and means with standard deviations (SD) for continuous variables. To examine bivariate associations between key variables, independent samples t-tests or one-way ANOVA were used for continuous outcomes, and chi-square tests were applied for categorical outcomes.

Multivariable linear regression models were then fitted to evaluate the associations between independent predictors and the dependent outcome variables, while controlling for potential confounders such as gender, age, and employment status (enter method). Regression assumptions (normality, homoscedasticity, and multicollinearity) were checked and met.

Furthermore, Path analysis—as a specific form of Structural equation modeling (SEM)—was conducted using Maximum likelihood estimation to test the hypothesized mediation model. Specifically, the mediating role of resilience in the association between social support (independent continuous variable) and quality of life (QoL, dependent continuous variable) was examined. The significance of indirect effects was assessed using the Bootstrapping method with 5,000 resamples to obtain bias-corrected 95% confidence intervals (CI). The proportion of the total effect of social support on QoL explained by the mediation pathway was computed by dividing the indirect effect by the total effect. All analyses were conducted at a 95% CI and a two-sided significance level of $p < 0.05$ using Stata version 14.2 (StataCorp, College Station, TX, USA).

### Results

### Basic characteristics of the participants

According to the results of Table 1, most of the participants were male (72.22%), in the age range of 65–75 years (46.2%) and retired (49.2%).

**Table 1. Demographic characteristics of participants.**

| Variables | | Frequency (%) |
|---|---|---|
| Gender | Male | 247 (72.22) |
| | Female | 95 (27.78) |
| Age | 60-65 | 73 (21.35) |
| | 65-70 | 76 (22.22) |
| | 70-75 | 82 (23.98) |
| | 75-80 | 63 (18.42) |
| | 80-85 | 38 (11.11) |
| | >85 | 10 (2.92) |
| Job status | Retired | 169 (49.42) |
| | Unemployment | 167 (49.83) |
| | Employed | 6 (1.75) |

As shown in Table 2, mean score of resilience and social support in men was 59.56 out of 125 and 80.11 out of 115, respectively, and more than women. Meanwhile, mean score of the HRQOL in women was more than men (94.23 out of 100). Also, with increasing age, the mean score of all three variables has decreased. The mean score of resilience was lower and the mean score of HRQOL in employed individuals was higher than others. Social support was more in the unemployed group than others.

### Testing research hypotheses

**Relationship between social support and resilience.** As shown in Table 3, after adjusting for gender, age, and job status, a one-unit increase in social support is associated with a 0.44 increase in resilience on average ($\beta = 0.44$, $p = 0.001$).

**Relationship between social support and HRQOL.** As shown in Table 3, after controlling for gender, age, and job status, a one-unit increase in social support is associated with a 0.24 increase in HRQoL on average ($\beta = 0.24$, $p = 0.001$).

**Table 2. Descriptive analysis of main variables based on demographic characteristics.**

| Demographic characteristics | | Variables | | | | | |
|---|---|---|---|---|---|---|---|
| | | Resilience | | Social support | | Quality of life | |
| | | Mean | SD | Mean | SD | Mean | SD |
| Gender | Female | 54.77 | 11.8 | 73.69 | 10.35 | 94.23 | 7.44 |
| | Male | 59.56 | 10.84 | 80.11 | 12.42 | 92.72 | 10.16 |
| Age | 60-65 | 61.91 | 11.15 | 81.67 | 13.09 | 93.37 | 10.46 |
| | 65-70 | 60.5 | 10.94 | 79.61 | 11.64 | 93.47 | 10.1 |
| | 70-75 | 57.79 | 9.26 | 77.51 | 11.04 | 93.68 | 9.46 |
| | 75-80 | 55.65 | 11.37 | 76.74 | 10.54 | 93.22 | 7.54 |
| | 80-85 | 52.5 | 10.15 | 74.1 | 12.9 | 91.73 | 8.6 |
| | >85 | 55.9 | 20.5 | 77 | 19.99 | 89.3 | 12.74 |
| Job status | Employed | 57.37 | 11.31 | 76.37 | 11.82 | 93.45 | 8.78 |
| | Retired | 57.26 | 11.18 | 81.05 | 11.54 | 93.37 | 9.92 |
| | Unemployment | 53.16 | 3.65 | 66.16 | 11.77 | 94.5 | 6.25 |
| Total | | 58.23 | 11.31 | 78.33 | 12.21 | 93.14 | 9.5 |

**Table 3.  Linear regression analysis assessing the relationships between social support, resilience, and HRQoL adjusted for gender, age, and job status.**

| Variables | Resilienc | | | | | HRQoL | | | | |
|---|---|---|---|---|---|---|---|---|---|---|
| | Coef. | SE | T-value | P-value | CI | Coef. | SE | T-value | P-value | CI |
| Social support | 0.44 | 0.04 | 10.19 | 0.001 | 0.35,0.53 | 0.24 | 0.04 | 5.85 | 0.001 | 0.85, 5.31 |
| Resilience | | | | | | 0.34 | 0.04 | 7.66 | 0.001 | 0.25, 0.42 |

**Relationship between resilience and HRQOL.**  As shown in Table 3, after adjusting for gender, age, and job status, a one-unit increase in resilience is associated with a 0.34 increase in HRQoL on average (β = 0.34, p = 0.001).

**Relationship between social support, resilience and HRQOL.**  As shown in Table 4, resilience and social support simultaneously explain the HRQoL in participants.

The research hypothesis of our study is displayed in Fig 1. In the research hypothesis, a mediation model was fitted where social support was set as independent variable, resilience as a mediator, and quality of life as dependent variable. Results shows that the social support had a significant direct effect (β = 0.097) and an indirect effect through resilience (β = 0.12) on the quality of life. We also found a significant total effect of social support (β = 0.22) on the quality of life (Table 5). The model showed that a proportion of 55.8% of the total effect is explained by the mediator.

**Table 4.  Linear regression analysis assessing the relationships between social support, resilience and HRQoL, simultaneously.**

| Variables | HRQoL | | | | |
|---|---|---|---|---|---|
| | Coef. | SE* | T-value | P-value | CI |
| Resilienc | 0.27 | 0.05 | 5.46 | 0.000 | 0.17, 0.37 |
| Social Support | 0.12 | 0.04 | 2.68 | 0.008 | 0.03, 0.21 |
| Gender | 3.64 | 1.09 | 3.34 | 0.001 | 1.49, 5.79 |
| Age | 0.33 | 0.35 | 0.96 | 0.33 | − 0.35,1.02 |
| Retired | −0.16 | 0.43 | − 0.39 | 0.69 | − 01.01, 0.67 |

*SE: Standard Error.

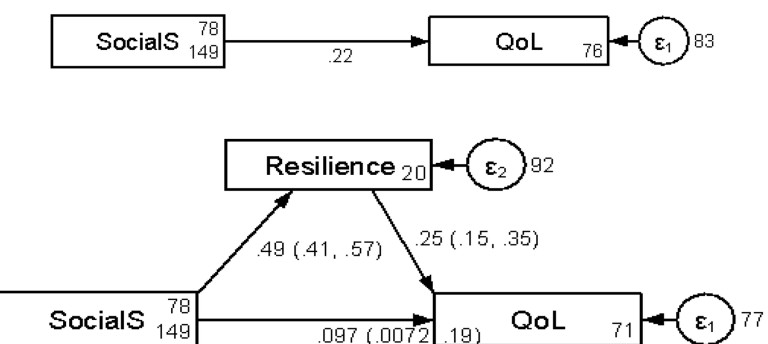

**Fig 1.  Pathways between, social support, resilience, and Health related Quality of life (*Social S: social support, QoL: Health Related Quality of life).**

**Table 5. Direct, indirect, and total effects of social support on HRQoL.**

| Pathways | Coefficient | SE* | P-value |
|---|---|---|---|
| **Direct effects** | | | |
| Resilience →HRQoL | 0.25 | 0.05 | 0.001 |
| Social support →HRQoL | 0.097 | 0.05 | 0.034 |
| Social support→Resilience | 0.49 | 0.04 | 0.001 |
| **Indirect effects** | | | |
| Social support →HRQoL | 0.12 | 0.03 | 0.001 |
| **Total effects** | | | |
| Social support →HRQoL | 0.22 | 0.04 | 0.001 |

*SE: Standard Error.

## Discussion

### Main findings

In this specific cross-sectional study, the focus was on examining the mediating role of resilience in the relationship between social support and HRQoL among elderly during Covid-19 pandemic. The results indicated that the availability of appropriate social support is a key factor in improving both resilience and HRQoL. Notably, these effects remained significant even after accounting for potential confounding variables such as demographic characteristics.

### Available evidence on the association of social support and resilience with HRQoL

Existing literature provides insight into correlation between social support, resilience, and HRQoL. Our analysis revealed a direct impact of social support on HRQoL amounting to 0.44 ($p < 0.001$). Prior research has illustrated the capacity of social support to influence HRQoL during the Covid-19 outbreak [17,14,16]. Noteworthy is the intervention research conducted by Ajh et al., which underscored the nexus between social support and the enhancement of elderly individuals' quality of life, resulting in a noticeable augmentation in both mental and physical dimensions post-intervention [38]. Additionally, antecedent studies indicate that social elements such as social engagement, familial affiliation, level of familial reverence for seniors, amicable ties with neighbors, contentment with residential environment, participation in communal activities, and satisfaction derived from interpersonal interactions significantly impact their overall quality of life [39,40].

In this investigation, a positive and statistically significant correlation between resilience and quality of life has been validated, aligning with previous findings by Fam [41], Gerino [42], and Silverman [43]. The outcomes highlight resilience as a protective element, capable of enhancing quality of life through mitigation of adverse impacts stemming from unfavorable circumstances. Furthermore, they underscore the significance of assessing an individual's capacity to navigate challenging situations, thereby influencing quality of life [44]. Consequently, the identification of older adults most vulnerable to detrimental effects of the pandemic on quality of life is crucial, necessitating the development of a tailored health intervention.

Furthermore, the present study results demonstrate that both resilience and social support serve as significant predictors of HRQoL among elderly during Covid-19 pandemic. These findings are in agreement with the studies of Mikayili et al., [45], Salonenet et al., [46], Behnam Moghadam et al., [47], Patel et al., [48], Zhou et al., [49] which conducted before the Covid-19 outbreak. According to the results of our study, social support has a moderate direct effect on the resilience of the elderly ($\beta = 0.49$). However, the direct effect of social support on HRQoL is weaker than the direct effect of resilience. This means that to improve the HRQoL of the elderly, planning and interventions should focus on improving the resilience of the elderly.

In our study, effect of resilience on HRQOL (as mediator variable) was moderate effect (β = 0.49). This finding is not in line with the results of Kong's et al., study among migrant older adults prior to the onset of the Covid-19 pandemic [50]. The findings derived from the mentioned study conducted demonstrated that resilience served as a partial mediator in the association between social support and both dimensions of HRQOL. These results enhance our comprehension of the interplay between social support and psychological resilience in shaping HRQOL, thereby offering valuable insights for designing interventions aimed at improving HRQOL among older adults residing in community settings [50].

In the research conducted by Wu et al., [51] encompassing a sample of 205 elderly individuals residing in nursing homes, the effect of social support on HRQOL was stronger than that effect in our study (β = 0.30, p < 0.001), but the effect of resilience as mediating role in the relationship between social support and HRQOL was smaller (β = 0.13, p = 0.008). To justify these findings, it can be stated that elderly individuals residing in nursing homes obtain more substantial social support from their caregivers than those living in the community. However, this social support does not significantly enhance resilience among these individuals. It is essential that this support is structured to bolster the elderly's ability to adapt and confront the challenges associated with living in nursing homes, thereby improving resilience and ultimately enhancing their HRQOL. It is also necessary to identify other variables that can have a stronger effect on the relationship between social support and quality of life in these individuals, so that effective interventions can be taken to improve the HRQoL of these individuals.

Finally, individuals with higher levels of resilience exhibit increased satisfaction and quality of life through the attenuation of negative emotions and fostering positive interpretations of experiences. Moreover, those with greater social support tend to enjoy enhanced quality of life, with social support playing a role in bolstering mental well-being by alleviating stress and anxiety. Consequently, the enhancement of resilience and social support is conducive to improving HRQoL [52].

## Strengths and limitations

This research exhibited several notable strengths. Initially, it identified the correlation between social assistance and HRQoL among the elderly populace amid the Covid-19 crisis. Furthermore, it scrutinized the intermediary function of resilience in the correlation between social support and HRQoL. The outcomes of this study hold the potential to guide the focal point of strategizing for policymakers and administrators. The study encompassed 341 elderly individuals residing within the community, thereby enhancing the external validity of our findings within the Iranian context. Nonetheless, it is imperative to acknowledge that HRQoL is influenced by a multitude of factors including place of habitation, cultural disparities, and various socio-economic aspects. To address these considerations, research participants should be recruited from diverse urban settings, while controlling for these variables to scrutinize the interrelation of the variables under study. It is advisable for future researchers to explore other conceivably influential factors in the association between social support and HRQoL, bearing in mind the aforementioned aspects for subsequent investigations.

## Conclusion

Drawing from the findings of this investigation, resilience emerged as a pivotal mediator in the nexus between social support and HRQoL among the elderly during the Covid-19 pandemic, underscoring the necessity to enhance the HRQoL of older individuals during crises, particularly within aging and elderly communities. In addition to emphasizing social support, enhancing resilience through imparting skills such as problem-solving and self-awareness is recommended to enhance the quality of life for this demographic, especially during their later years. In addition, through the implementation of spiritual therapy techniques, facilitating participation in group sessions focused on spiritual therapy, and fostering the exchange of thoughts and ideas, policymakers in the aging sector can enhance the capacity and consciousness of the elderly, reinforcing the notion that life invariably holds significance and that meaning and purpose can be discovered in every experience.

## Supporting information

**S1 File. Data underlying the findings described in our manuscript.**
(PDF)

## Author contributions

**Conceptualization:** Badriyeh Karami, Mansour Bagherinia, Ebrahim Shakiba, Raheleh Maleki, Hadi Darvishigilan.

**Data curation:** Shahab Rezaeian, Mansour Bagherinia, Mohammadrasool Ghasemianrad, Amirhossien Naghibzadeh, Hamzeh Zahabi.

**Formal analysis:** Shahab Rezaeian.

**Writing – original draft:** Badriyeh Karami, Mansour Bagherinia, Hadi Darvishigilan.

**Writing – review & editing:** Badriyeh Karami, Mansour Bagherinia, Hadi Darvishigilan.

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
