## [Decision Letter · Decision Letter 0]

8 Sep 2025

PONE-D-24-51857Social Support and Health Related Quality of Life among Older People in Covid-19 Pandemic: The Mediating Role of ResiliencePLOS ONE

Dear Dr. Darvishigilan,

Thank you for submitting your manuscript to PLOS ONE. After careful consideration, we feel that it has merit but does not fully meet PLOS ONE’s publication criteria as it currently stands. Therefore, we invite you to submit a revised version of the manuscript that addresses the points raised during the review process.

We look forward to receiving your revised manuscript.

Kind regards,

Bibi Razieh Hosseini Farash

Academic Editor

PLOS ONE

Journal Requirements:

2. We note that your Data Availability Statement is currently as follows: All relevant data are within the manuscript and in Supporting Information files.

Reviewers' comments:

Reviewer's Responses to Questions

**Comments to the Author**

1. Is the manuscript technically sound, and do the data support the conclusions?

Reviewer #1: Yes

Reviewer #2: Partly

2. Has the statistical analysis been performed appropriately and rigorously? 

Reviewer #1: No

Reviewer #2: Yes

3. Have the authors made all data underlying the findings in their manuscript fully available?

Reviewer #1: Yes

Reviewer #2: Yes

4. Is the manuscript presented in an intelligible fashion and written in standard English?

Reviewer #1: Yes

Reviewer #2: Yes

5. Review Comments to the Author

Reviewer #1: 1. If the basis for dividing regions into weak, good, and strong categories is mentioned in the text of the article, specifying the criteria for these divisions (such as income, education, etc.) can enhance the reader's understanding.

2. Additionally, the data analysis should be presented in a more detailed manner, for example, stating how reliability and validity coefficients were calculated."

3. Enhance Statistical Analysis: Providing detailed descriptions of the statistical methods used and presenting comprehensive analysis results would increase transparency and scientific validity.

4. Improve Results Structure: Organizing and structuring the results section more coherently would enhance readability and comprehension.

Reviewer #2: Dear Editor

Thank you for the opportunity to review. The review of manuscript PONE-D-24-51857, entitled “Social Support and Health Related Quality of Life among Older People in Covid-19 Pandemic: The Mediating Role of Resilience” done and sent for you.

Comments to authors

Abstract

- Several grammatical errors are present, such as line 42 "341 older people over 60 years were participated," which should be corrected to "341 older adults over 60 years participated".

- In line 39, Phrasing such as "it will be possible to improve…" is vague and speculative. A more assertive tone is recommended.

- In line 48, there are misplaced commas (e.g., β = 0.44,).

- In lines 48 and 49, the statistical findings are not contextualized—what does a β = 0.097 mean in practical terms for HRQoL?

- I recommended briefly mention prior studies and highlight the novelty of this research.

Introduction

- I would suggest to summarize the introduction section.

- Definitions and relationships between HRQoL and social support are repeated unnecessarily.

- The concept of resilience is introduced multiple times with overlapping definitions and citations, which disrupts the flow and dilutes impact.

- The paragraph transitions are abrupt, and the logical progression of ideas is weak. For example, demographic data about Iran’s aging population is inserted without a clear link to the preceding or following content.

- The final paragraph claims novelty in examining the three variables concurrently but does not adequately support this claim with references or a synthesis of existing gaps in the literature.

In line 100, the writing is overly formal and verbose in places, which may hinder readability. Phrases such as "engendered by the emergence of this recent epidemic" could be simplified.

Materials and Methods

- Please explain the demographic factors.

- In line 137, Please replace the word "Instrument" in this sentence with "The questionnaires".

- Authors should be summarized the parts mentioned in lines 138-169.

- The study uses cluster random sampling across 16 neighborhoods, but the rationale for selecting two neighborhoods per district is not fully explained.

- It is recommended to clarify how the sample size was determined and whether power analysis was conducted to justify the inclusion of 341 participants.

- The method of replacing non-respondents with others from the same age group and neighborhood may introduce selection bias so, it is recommended to discuss the potential impact of this substitution on sample representativeness and data validity.

- While reliability coefficients are reported for all instruments, the validity of the resilience questionnaire in the elderly population is not discussed. It is recommended to provide more detail on the cultural and age-specific validation of the Conner-Davidson resilience scale in Iranian elderly populations.

- The authors mention controlling for gender, age, and employment status, but do not specify how these were included in the regression or SEM models.

- Include a statement confirming ethical clearance from an institutional review board and the process for obtaining participant consent.

- Clarify whether missing data were present and how they were handled.

Results

- In lines 185 and 188, tables and 2 change to Tables 1 and 2.

- In line 206, the relationship between resilience and HRQoL is reported with two different β values (0.34 and 0.24), which creates confusion.

- Differences in mean scores between demographic groups (e.g., gender, employment status, age) are described, but no statistical tests (e.g., t-tests, ANOVA) are reported to determine significance.

- The scales used for resilience (out of 125), social support (out of 115), and HRQoL (out of 100) are mentioned, but the instruments are not identified.

- The phrase "considering demographic variables" is used repeatedly, but it is unclear which variables were controlled for and how they were incorporated into the models.

- Some reported β values (e.g., 0.097 for direct effect) are statistically significant but small. The practical relevance of these findings is not discussed.

- The finding that unemployed individuals had higher social support than employed ones contradicts common assumptions and prior literature.

Discussion

- The reported direct effect of social support on HRQoL (β = 0.04) is statistically significant but extremely small. The discussion does not address whether this effect is practically meaningful.

- While the authors cite numerous studies, the discussion lacks critical analysis of conflicting findings, methodological differences, or contextual limitations.

- The mediating role of resilience is reiterated multiple times without adding new insights or elaborating on mechanisms. It is recommended to consolidate repetitive statements and expand on how resilience functions as a mediator in practical terms.

- The manuscript suggests that findings can inform policy but does not specify how interventions might be designed or implemented. Offer concrete examples of resilience-building or social support strategies that could be applied in community or clinical settings.

References

- Please correct the errors in throughout the References. However, Authors must use the abbreviation for journal name. Articles rewrite according to "PLOS ONE" format.

- In line 355, "WH" change "WHO".

- Please delete "" in line 377.

- In lines 406 and 409, articles with more than 6 authors must also rewrite according to "PLOS ONE" format.

- In line 423, please delete ".", ":" and ";".

- In line 424, Karimi Y KM HY should be corrected.

- 2006;5(1):0-. in line 431, rewrite.

- Please delete "," in line 434.

- In line 444, authors should be corrected.

- In line 450, please delete space and ",".

- In lines 466 and 472, authors should be corrected.

Tables

- Authors should explain "*" in Table 5.

6. PLOS authors have the option to publish the peer review history of their article (what does this mean?). If published, this will include your full peer review and any attached files.

Reviewer #1: No

Reviewer #2: No

---

## [Author Response · Author response to Decision Letter 1]

27 Sep 2025

respose to reviewers comment was uploaded.

ID number: PONE-D-24-51857

Dear editor of journal of PLOS One

Thanks for providing the comments of the respectful reviewer to us. We tried to revise the manuscript, titled “Social Support and Health Related Quality of Life among Older People in Covid-19 Pandemic: The Mediating Role of Resilience”, based on the comments and respond them in the following table. Revisions has been highlighted in green color in the manuscript. Hope the revisions are satisfactory. We welcome any further constructive comments if required.

Dr. Hadi Darvishigilan

Corresponding author

Comment Response

Reviewer #1

1. If the basis for dividing regions into weak, good, and strong categories is mentioned in the text of the article, specifying the criteria for these divisions (such as income, education, etc.) can enhance the reader's understanding. Many thanks for your comment. More explanation was added to the material and methods; Sample and sampling method section. Pages 4 and 5, lines 119-124.

2. Additionally, the data analysis should be presented in a more detailed manner, for example, stating how reliability and validity coefficients were calculated." Thank you for your comment. The questionnaires used in this study have been previously validated and shown to have good reliability in prior research. Therefore, we relied on these established instruments for measuring the constructs in our study. We have added these details and references to the material and methods; questionnaire section to clarify the reliability and validity of the measures. Pages 5 and 6, lines 132-156.

More explanation about data analysis was presented in a more detailed manner in material and methods; statistical analysis section. Pages 6 and 7, pages 168-186.

3. Enhance Statistical Analysis: Providing detailed descriptions of the statistical methods used and presenting comprehensive analysis results would increase transparency and scientific validity. We have clarified this point in the material and methods; statistical analysis section of the revised manuscript. Pages 6 and 7, pages 168-186.

4. Improve Results Structure: Organizing and structuring the results section more coherently would enhance readability and comprehension. Thank you. It was done. Pages 7 and 8, lines 188- 223.

Reviewer #2

Abstract

1. Several grammatical errors are present, such as line 42 "341 older people over 60 years were participated," which should be corrected to "341 older adults over 60 years participated". Many thanks for your comment. Grammatical errors were corrected. Page 2, line 43.

2. In line 39, Phrasing such as "it will be possible to improve…" is vague and speculative. A more assertive tone is recommended. Many thanks for your comment. The mentioned sentence was revised. Page 2, lines 39-40.

3. In line 48, there are misplaced commas (e.g., β=0.44,). Many thanks for your attention. It was deleted.

Page 2, line 49.

4. In lines 48 and 49, the statistical findings are not contextualized—what does a β = 0.097 mean in practical terms for HRQoL? Many thanks for your comment. More explanation was added to the result section of the abstract. Page 2, lines 48-52.

Introduction

1. I would suggest to summarize the introduction section. Many thanks for your comment. The introduction section was revised. Pages 3 and 4, lines 65-107.

2. Definitions and relationships between HRQoL and social support are repeated unnecessarily. Many thanks for your comment. The introduction section was revised. Pages 3 and 4, lines 65-107.

3. The concept of resilience is introduced multiple times with overlapping definitions and citations, which disrupts the flow and dilutes impact. Many thanks for your comment. The introduction section was revised. Pages 3 and 4, lines 65-107.

4. The paragraph transitions are abrupt, and the logical progression of ideas is weak. For example, demographic data about Iran’s aging population is inserted without a clear link to the preceding or following content. Many thanks for your comment. The introduction section was revised. Pages 3 and 4, lines 65-107.

5. The final paragraph claims novelty in examining the three variables concurrently but does not adequately support this claim with references or a synthesis of existing gaps in the literature. Many thanks for your comment. The introduction section was revised. Pages 3 and 4, lines 65-107.

6. In line 100, the writing is overly formal and verbose in places, which may hinder readability. Phrases such as "engendered by the emergence of this recent epidemic" could be simplified. Many thanks for your comment. The introduction section was revised. Pages 3 and 4, lines 65-107.

7. I recommended briefly mention prior studies and highlight the novelty of this research. Many thanks for your comment. The introduction section was revised. Pages 3 and 4, lines 65-107.

Materials and Methods

1. Please explain the demographic factors. Many thanks for your comment. More explanations was added to the material and methods; Study setting and Sample and sampling method section. Page 4, line 111. Page 5, lines 128-130.

2. In line 137, Please replace the word "Instrument" in this sentence with "The questionnaires". Many thanks for your comment. It was edited. Page 5, line 132.

3. Authors should be summarized the parts mentioned in lines 138-169. Many thanks for your comment. This part was summarized. Pages 5 and 6, lines 132-156.

4. The study uses cluster random sampling across 16 neighborhoods, but the rationale for selecting two neighborhoods per district is not fully explained. Many thanks for your comment. More explanation was added to the material and methods; Sample and sampling method section. Pages 4 and 5, lines 119-124.

5. It is recommended to clarify how the sample size was determined and whether power analysis was conducted to justify the inclusion of 341 participants. Many thanks for your comment. The procedure for determining the sample size was described comprehensively. Page 4, lines 115-119.

6. The method of replacing non-respondents with others from the same age group and neighborhood may introduce selection bias so, it is recommended to discuss the potential impact of this substitution on sample representativeness and data validity. Many thanks for your comment. The sampling method was explained more clearly in the method section. Pages 4 and 5, lines 115-130. Given that the participants were community-dwelling seniors aged over 60 years, the interviewers accessed the samples by visiting parks and recreational areas in the neighborhoods, and this process continued until the total number of questionnaires was completed, there is no selection bias.

7. While reliability coefficients are reported for all instruments, the validity of the resilience questionnaire in the elderly population is not discussed. It is recommended to provide more detail on the cultural and age-specific validation of the Conner-Davidson resilience scale in Iranian elderly populations. Thank you for your comment. The questionnaires used in this study have been previously validated and shown to have good reliability in prior research [1, 2]. Therefore, we relied on these established instruments for measuring the constructs in our study. We have added these details and references to the methods section to clarify the reliability and validity of the measures.

More explanation added to the methods; questionnaire section. Pages 5 and 6, lines 132-156.

8. The authors mention controlling for gender, age, and employment status, but do not specify how these were included in the regression or SEM models. We appreciate the reviewer’s valuable comment. In our regression analyses, the covariates (gender, age, and employment status) were included as control variables using the enter method, meaning they were entered simultaneously in a single block along with the main independent variables. This approach allowed us to adjust for their potential confounding effects on the outcome variable.

9. Include a statement confirming ethical clearance from an institutional review board and the process for obtaining participant consent. In the method; data collection and declaration sections, the process of obtaining the ethics code from Kermanshah University of Medical Sciences is mentioned, and the ethics code is also included.

Page 6, lines 158-166.

Page 12, lines 324-329.

10. Clarify whether missing data were present and how they were handled. We appreciate the reviewer’s comment. There were no missing data in our dataset; therefore, no imputation or special handling procedures were required. We have clarified this point in the Methods section of the revised manuscript.

Results

1. In lines 185 and 188, tables and 2 change to Tables 1 and 2. Many thanks for your attention. It was edited.

Page 7, lines 190-193.

2. In line 206, the relationship between resilience and HRQoL is reported with two different β values (0.34 and 0.24), which creates confusion. Many thanks for your attention. It was corrected. Page 8, lines 209-211.

3. Differences in mean scores between demographic groups (e.g., gender, employment status, age) are described, but no statistical tests (e.g., t-tests, ANOVA) are reported to determine significance. Thank you for your comment. The table was intended to provide a descriptive overview of mean differences across demographic groups. Inferential statistical analyses (e.g., Linear regression) was conducted in subsequent analyses, and the results are reported in Table 3 and 4. Page 18.

4. The scales used for resilience (out of 125), social support (out of 115), and HRQoL (out of 100) are mentioned, but the instruments are not identified. Many thanks for your attention. Detailed explanations of the questionnaires used are provided in the method section, and the score obtained from each questionnaire is stated in the Results section. Pages 5 and 6. Lines 132-156.

5. The phrase "considering demographic variables" is used repeatedly, but it is unclear which variables were controlled for and how they were incorporated into the models. Thank you for your comment. We have clarified which demographic variables were included in the analyses. Page 6. Lines 174-177.

Specifically, gender, age, and job status were controlled for in all models. Accordingly, the text in the Results section has been revised to specify that the reported associations are adjusted for these variables. Pages 7 and 8. Lines 200-211.

6. Some reported β values (e.g., 0.097 for direct effect) are statistically significant but small. The practical relevance of these findings is not discussed. The Results section has been revised. The Discussion section has been revised and the interpretation of the results is provided in the Discussion section.

7. The finding that unemployed individuals had higher social support than employed ones contradicts common assumptions and prior literature. After reviewing the research findings by the study's statistical consultant, it was determined that an error had occurred in entering the information in the relevant table. The findings were reviewed and the correct finding was entered in Table 2. Page 17.

Discussion

1. The reported direct effect of social support on HRQoL (β = 0.04) is statistically significant but extremely small. The discussion does not address whether this effect is practically meaningful. Thank you for your comment. The discussion section was revised based on the comments of the respected reviewers. Pages 8, 9, and 10. Lines 227-288.

2. While the authors cite numerous studies, the discussion lacks critical analysis of conflicting findings, methodological differences, or contextual limitations. Thank you for your comment. The discussion section was revised based on the comments of the respected reviewers. Pages 8, 9, and 10. Lines 227-288.

3. The mediating role of resilience is reiterated multiple times without adding new insights or elaborating on mechanisms. It is recommended to consolidate repetitive statements and expand on how resilience functions as a mediator in practical terms Thank you for your comment. The discussion section was revised based on the comments of the respected reviewers. Pages 8, 9, and 10. Lines 227-288.

4. The manuscript suggests that findings can inform policy but does not specify how interventions might be designed or implemented. Offer concrete examples of resilience-building or social support strategies that could be applied in community or clinical settings. Thank you for your comment. The discussion section was revised based on the comments of the respected reviewers. Pages 8, 9, and 10. Lines 227-288.

References

1. Please correct the errors in throughout the References. However, Authors must use the abbreviation for journal name. Articles rewrite according to "PLOS ONE" format. The PLOS One journal reference style has been followed.

2. In line 355, "WH" change "WHO". This reference was deleted in the revised manuscript.

3. Please delete "" in line 377. It was corrected.

4. In lines 406 and 409, articles with more than 6 authors must also rewrite according to "PLOS ONE" format. The PLOS One journal reference style has been followed.

5. In line 423, please delete ".", ":" and ";". It was edited.

6. In line 424, Karimi Y KM HY should be corrected. It was edited.

7. 2006;5(1):0-. in line 431, rewrite. It was edited.

8. Please delete "," in line 434. It was edited.

9. In line 444, authors should be corrected. It was edited.

10. In line 450, please delete space and ",". It was edited.

11. In lines 466 and 472, authors should be corrected It was edited.

Tables

1. Authors should explain "*" in Table 5. It was added.

1. Noroozi Homayoon M, Sadeghi M, Sadri Damirchi E, Hatami Nejad M. The relationship between sense of coherence and perceived social support with resilience: The mediating role of adaptive cognitive-emotional regulation strategies in older adults. Aging Psychology. 2024;10(4):427-05.

2. Yousefi Afrashteh M, Moradi M. The Spiritual Health and Social Support with Quality of Life in the Rural Elderly: The Mediating Role of Resilience and Hope. Aging Psychology. 2022;8(3):266-51.

---

## [Decision Letter · Decision Letter 1]

2 Feb 2026

Social Support and Health Related Quality of Life among Older People in Covid-19 Pandemic: The Mediating Role of Resilience

PONE-D-24-51857R1

Dear Dr. Hadi Darvishi,

We’re pleased to inform you that your manuscript has been judged scientifically suitable for publication and will be formally accepted for publication once it meets all outstanding technical requirements.

Kind regards,

Bibi Razieh Hosseini Farash

Academic Editor

PLOS One

Additional Editor Comments (optional):

Reviewers' comments:

Reviewer's Responses to Questions

**Comments to the Author**

1. If the authors have adequately addressed your comments raised in a previous round of review and you feel that this manuscript is now acceptable for publication, you may indicate that here to bypass the “Comments to the Author” section, enter your conflict of interest statement in the “Confidential to Editor” section, and submit your "Accept" recommendation.

Reviewer #2: All comments have been addressed

Reviewer #3: All comments have been addressed

2. Is the manuscript technically sound, and do the data support the conclusions?

Reviewer #2: Yes

Reviewer #3: Yes

3. Has the statistical analysis been performed appropriately and rigorously? 

Reviewer #2: Yes

Reviewer #3: Yes

4. Have the authors made all data underlying the findings in their manuscript fully available?

Reviewer #2: Yes

Reviewer #3: No

5. Is the manuscript presented in an intelligible fashion and written in standard English?

Reviewer #2: Yes

Reviewer #3: Yes

6. Review Comments to the Author

Reviewer #2: Dear Authors

Some minor corrections need to be made in manuscript as follows:

Page 2, line 67: "the advent of epidemics" can be replaced to "the onset of epidemics"

Page 2, line 68: "While the disease can impact" can be replaced to "While the disease can affect"

Page 2, line 70: "This matter holds significant importance for nations experiencing population aging. Iran is one of the countries with a high rate of population aging [4]. Iran is poised to witness a substantial rise in the proportion of its elderly population over the next three decades [5]." Should be replaced by "This issue is particularly important for nations experiencing population aging. Iran is one of the countries with a high rate of population aging [4] and is expected to see substantial increase in the proportion of its elderly population over the next three decades [5].

Page 4, Line 111: carried out

Pages 8, 9, lines 228-233: In this specific cross-sectional study, the focus was on examining the mediating role of resilience in the relationship between social support and HRQoL among the elderly during the Covid-19 pandemic. The results indicated that the availability of appropriate social support is a key factor in improving both resilience and HRQoL. Notably, these effects remained significant even after accounting for potential confounding variables such as demographic characteristics.

Page 9, lines 236: Existing literature provides insight into...

Reviewer #3: Thank you for your cooperation and kindness . All the points I had suggested have been corrected.

Recommendations:

1. If the basis for dividing regions into weak, good, and strong categories is mentioned in the text of the article, specifying the criteria for these divisions (such as income, education, etc.) can enhance the reader's understanding.

2. Additionally, the data analysis should be presented in a more detailed manner, for example, stating how reliability and validity coefficients were calculated."

3. Enhance Statistical Analysis: Providing detailed descriptions of the statistical methods used and presenting comprehensive analysis results would increase transparency and scientific validity.

4. Improve Results Structure: Organizing and structuring the results section more coherently would enhance readability and comprehension.

7. PLOS authors have the option to publish the peer review history of their article (what does this mean?). If published, this will include your full peer review and any attached files.

Reviewer #2: No

Reviewer #3: No

---

## [Editor Report · Acceptance letter]

PONE-D-24-51857R1

PLOS One

Dear Dr. Darvishigilan,

I'm pleased to inform you that your manuscript has been deemed suitable for publication in PLOS One. Congratulations! Your manuscript is now being handed over to our production team.

Kind regards,

on behalf of

Dr. Bibi Razieh Hosseini Farash

Academic Editor

PLOS One